# Seroprevalence of *Toxoplasma gondii* in Cats from Cyprus

**DOI:** 10.3390/pathogens10070882

**Published:** 2021-07-12

**Authors:** Charalampos Attipa, Christos Yiapanis, Séverine Tasker, Anastasia Diakou

**Affiliations:** 1Department of Tropical Disease Biology, Liverpool School of Tropical Medicine, Liverpool L3 5QA, UK; 2Malawi Liverpool Wellcome Trust Clinical Research Programme, Blantyre, Malawi; 3Cyvets Veterinary Center, Paphos 8025, Cyprus; drcy@cyvets.com; 4Bristol Veterinary School, University of Bristol, Bristol BS40 5DU, UK; S.Tasker@bristol.ac.uk; 5Linnaeus Group, Shirley, Solihull B90 4BN, UK; 6Faculty of Health Sciences, School of Veterinary Medicine, Aristotle University of Thessaloniki, Thessaloniki 54124, Greece

**Keywords:** Cyprus, domestic cat, seroprevalence, *Toxoplasma gondii*

## Abstract

*Toxoplasma gondii* is a cosmopolitan protozoon parasite, and the causative agent of toxoplasmosis, one of the most prevalent zoonotic parasitic diseases. Cats, as definitive hosts, spread the parasite via their faeces, but this occurs only for a very short period in their life. Seropositivity in cats, although not associated with current shedding of the parasite, is indicative of the infection in a cat population and can be used to assess the infection risk for definitive and intermediate hosts in that area. In order to assess the prevalence of infection in cats living in Cyprus, 155 cats, originating from all districts of the country, were examined for the presence of *T*. *gondii* antibodies. Additionally, parameters such as age, sex, health status, lifestyle and concomitant infections were statistically assessed as potential risk factors for *T*. *gondii* seropositivity. Specific anti-*T*. *gondii* antibodies were detected in 50 (32.3%) cats, while the presence of feline immunodeficiency virus antibodies and a history of never having been vaccinated were statistically associated with *T*. *gondii* seropositivity on multivariate logistic regression analysis. This is the first report of *T*. *gondii* seroprevalence in cats in Cyprus and indicates that raised public awareness should be considered to prevent infection of animals and humans.

## 1. Introduction

The apicomplexan protozoon *Toxoplasma gondii* is one of the most extensively studied zoonotic parasites. It has a heteroxenous lifecycle that was fully described in 1970, when several different groups of investigators identified the domestic cat (*Felis catus*) as the definitive host [1]. The parasite passes its sexual, enteroepithelial phase in felids, and spreads into the environment in the form of environmentally resistant oocysts, shed through the faeces of infected felids. Sporulated oocysts, which develop from immature oocysts a few days after their excretion in cat faeces, are the infective stage for intermediate hosts, which comprise virtually all warm-blooded animals and humans [2,3]. Generally, both definitive and intermediate hosts may be infected (a) horizontally, by ingesting sporozoites in sporulated oocysts from the environment or by consuming bradyzoites and/or tachyzoites in tissue cysts in infected intermediate hosts, and (b) vertically, by transplacentally (or/and lactogenically in some species, e.g., goats, cats) transmitted tachyzoites [3,4].

*Toxoplasma gondii* infection usually remains subclinical in cats, although vertical toxoplasmosis or infection in immunocompromised cats may cause severe disease and even death [5]. Conversely, toxoplasmosis is considered one of the most important causes of abortion and, thus, of economic losses in livestock, particularly sheep and goat farming. Furthermore, toxoplasmosis has important zoonotic implications and infects, by estimation, nearly one third of the global human population [6], as one of the main foodborne diseases causing death [7].

At any given time, approximately 0.1–1% of cats are expected to shed oocysts, based on the observation that most cats only shed oocysts for about 1–2 weeks in their life [8,9]. However, in this short period of time, a cat can excrete millions of oocysts, which can remain infective in the environment for months and infect many hosts [10]. Diagnosis of infection in cats can be attempted by copromicroscopic examination; however this is not a helpful approach as (i) the enteroepithelial phase is very short, and (ii) other oocysts with indiscriminate morphology, i.e., species of the genera *Hammondia* and *Besnoitia*, can occur in cat faeces, necessitating molecular identification of the parasite [9].

Cats seroconvert approximately 1–2 weeks after the end of the enteroepithelial phase; thus, a seropositive cat no longer sheds oocysts [11]. Although seropositivity for *T. gondii* antibodies does not indicate that the cat is currently shedding the parasite, and not all seropositive cats have shed oocysts in the past [12,13], seroepizootiological studies are a useful indicative of infection in an area and a given cat population, and can be used to assess the infection risk for definitive and intermediate hosts in that same area [14]. 

Seroprevalance for *T. gondii* in domestic cats is estimated to be between 30 and 40%, worldwide [1]; however, depending on geography and sample size, seroprevalence has varied between 4.8% and 100.0% [5]. Although several reports on seroprevalence in cats have been published, there is still a paucity of information in many regions and countries of the world. The Republic of Cyprus is an islandic state in the eastern Mediterranean Sea, with a large number of cats, both pets and free-roaming or stray, that are often infected with multiple pathogens [15,16,17]. In this country, a limited number of surveys investigating the prevalence of *T*. *gondii* infection in humans and intermediate hosts have been reported [18,19]. However, the prevalence of infection in cats has never been investigated. In this context, the aim of the present study was to investigate the seroprevalence of *T*. *gondi* infection in cats from Cyprus, and to explore possible risk factors associated with seropositivity in cats. We report a seroprevalence of 32.3% of anti-*T*. *gondii* IgG in this population of cats and that a lack of vaccination and the presence of feline immunodeficiency virus (FIV) antibodies were associated with *T*. *gondii* seropositivity.

## 2. Materials and Methods

### 2.1. Study Design, Site Populations and Sampling

Blood samples of cats, both healthy and clinically ill, living in all six districts of Cyprus, i.e., Famagusta, Kyrenia, Larnaca, Limassol, Nicosia and Paphos, were collected in veterinary clinics from March to September 2014 as part of a previous study [17]. The samples were collected by the attending veterinarians during clinical routine examinations for healthy cats or medical checks for ill cats. Aliquots of 155 serum samples were used in the present study with the informed written consent of cat owners or animal shelter managers. For every cat, data concerning district of origin, habitat (rural or urban), age (≤1 year or >1 year, on the basis of health records), sex, breed, housing (access to outdoors or indoors only living), lifestyle (sheltered-stray or owned cat), previous travel history outside Cyprus, health status and vaccination status, use of ectoparasitic prevention and presence of anaemia (haematocrit <25%) based on in-house complete blood count were recorded. The serum samples were stored at −80 °C until they were transferred to the Laboratory of Parasitology and Parasitic Diseases, School of Veterinary Medicine, Aristotle University of Thessaloniki, Greece for examination.

### 2.2. Serological Examination

The serum samples were examined with the commercial kit ID Screen^®^ Toxoplasmosis Indirect Multi-species ELISA (ID.vet, Grabels, France), which detects anti-*T*. *gondii* IgG antibodies. The kit, which uses a purified peptide of the main P30 *T. gondii* protein as a substrate and a multi-species peroxidase (po) as conjugates, has shown a high level of concordance with other serological tests, and has been proved useful for toxoplasmosis serological screening in cats [20]. The test was performed following the manufacturer’s instructions. Briefly, for each sample, the sample to positive ratio (s/p %) was calculated taking the optical density (OD) of the positive control (pc) and negative control (nc) into account, as follows:$$\frac{s}{p}\;\%=\frac{\mathrm{ODs}-\mathrm{ODnc}}{\mathrm{ODpc}-\mathrm{ODnc}}\times 100$$

Samples with an s/p% ≤ 40% were considered negative, between 40% and 50% were considered doubtful and ≥50% were considered positive.

Additionally, in the framework of a previously published study [17], all samples underwent retroviral serology testing using PetCheck FeLV Antigen Test and PetCheck FIV Antibody Test (IDEXX Laboratories, Westbrook, ME, USA), *Leishmania infantum* antibody testing as well as PCR for *Bartonella henselae*, “*Candidatus* Mycoplasma haemominutum” (CMhm), “*Candidatus* Mycoplasma turicensis” (CMt) *Ehrlichia*/*Anaplasma* spp., *Hepatozoon* spp., *Leishmania* spp., and *Mycoplasma haemofelis* (Mhf).

### 2.3. Data Analysis

Univariate analysis was performed to evaluate for associations between seropositivity for *T*. *gondii* and demographic characteristics, as well as multiple co-infecting pathogens. Pearson’s Chi-square test was performed for the categorical variables. Variables with a trend towards significant association with *T*. *gondii* seropositivity (*p*-value < 0.2) were selected for further analysis with multivariate logistic regression. For the final model, a backward selection was used, and once a final model was structured, all the previously excluded variables were then individually retested and, if then significant, were included in the final model. For both the univariate and multivariate models, a *p*-value  ≤  0.05 was considered statistically significant. Statistical analysis was performed using SPSS for Windows (version 25.0; SPSS Inc., Chicago, IL, USA).

## 3. Results

Of the 155 examined samples, 50 (32.3%) were positive for *T*. *gondii* antibodies, with the remainder yielding negative results. At a district level, the percentage of seropositive cats was in Paphos 33.0%, Limassol 24.0%, Nicosia 36; 0%, Larnaca 31.0%, Kyrenia 40.0%, and Famagusta 29.0% (Figure 1). The age of seropositive cats ranged from 0.7 to 22.0 years (median 5.1 years, interquartile range 8 years), with 81 (52.3%) being male and 143 (92.3%) being non-pedigree cats. Table 1 and Table 2 show the demographic characteristics of this feline population, as well as the seroprevalence of *T*. *gondii* among the multiple co-infecting pathogens. The entire data set can be found in Appendix A. 

Univariate analysis (Table 1 and Table 2) revealed significant associations (*p* ≤ 0.05) between *T*. *gondii* seropositivity and the followings factors: never vaccinated, no application of ectoparasitic prevention, seropositivity for FIV, and positive *Hepatozoon* spp. PCR. Additionally, a trend toward significance (*p* < 0.2) for *T*. *gondii* seropositive was noted for cats that were non-pedigree, had outdoor access, lived in a rural area, were PCR-positive for CMhm and PCR-positive for at least one feline vector borne pathogen (FVBP).

A multivariate logistic regression model (Table 3) was constructed and yielded significant associations (*p* ≤ 0.05) for *T*. *gondii* seropositive that included FIV seropositivity (odds ratio (OR) = 5.0, 95% confidence interval (CI): 2.0–13.0, *p* = 0.001) and never being vaccinated (OR = 2.5, 95% CI: 1.2–5.7, *p* = 0.019). 

## 4. Discussion

*Toxoplasma gondii* is a prevalent parasite and toxoplasmosis affects a wide range of animals, both wild and domestic. Clinical toxoplasmosis in cats is rare; however it can occur in congenital infections that can be fatal, or postnatal infections in immunosuppressed cats, mainly involving the central nervous system and, less often, other organs, such as the lungs and eyes [21]. Farmed small ruminants, i.e., sheep and goats, are susceptible species, where toxoplasmosis is one the most important and prevalent causes of abortion, greatly affecting production and causing economic losses [22]. In Cyprus, the seroprevalence of *T*. *gondii* infection in small ruminants (sheep and goats) was found to be 40.1% in a recent survey [19].

Human toxoplasmosis in immunocompetent individuals usually remains asymptomatic, but rarely, it may cause a flu-like syndrome with fever, lymphadenopathy, hepatosplenomegaly and ocular lesions [23]. The most important implications of human toxoplasmosis are in immunosuppressed individuals, who can develop severe and even fatal disease, and in congenital infections that, depending on the gestational age at infection, can have a wide range of outcomes, from abortion to birth of asymptomatic infected infants [24,25]. In Cyprus, *T*. *gondii* seroprevalence in humans has been investigated in two groups of females: 16–18-year-old girls and pregnant women, showing prevalences of 6.5% and 18.0%, respectively [19]. A system for the epidemiological surveillance of symptomatic toxoplasmosis, both congenital or post-natal, has been employed in Cyprus since 2004 [26]. Nevertheless, there is evidence that such cases are still underreported [19].

In an epidemiological survey that took place in the early 2010s in several areas in Cyprus, Paphos displayed the highest seroprevalence in girls and women compared to the other districts of Cyprus [19]. However, in our study’s results, the seroprevalence in cats in Paphos was not higher than in other areas. This is not necessarily a contradictory observation, as the main mode of human infection by *T. gondii* is via consumption of raw or undercooked infected meat, or via ingestion of sporulated oocysts found on unwashed fruits and vegetables. Both ways of transmission involve foodstuffs that are moved from other areas of Cyprus, and are not related to contact with local contaminated soil [27]. In our present study, all districts of Cyprus displayed similar rates of feline infection, which is not surprising as there are no boundaries on the island that could isolate cat populations and, similarly, the intermediate hosts playing a key role in feline infection (rodents and birds) are abundant in all districts, and have been found to have a high percentage of up to 27.9% of seropositivity [18].

Cats are abundant in Cyprus, both as owned pets and as strays and, according to recent estimations, their population has reached 1.5 million [28]. Surveys investigating infectious and parasitic diseases of Cyprus cats are scant [15,17]. The current study is the first serological investigation of *T*. *gondii* infection in cats in Cyprus. Such surveys are needed to monitor the occurrence of this important pathogen in a given region and to predict the risk of infection in farm animals and humans [14].

In Europe, several *T*. *gondii* seroprevalence studies in feline populations have been conducted over the last decade [5]. The result of the present survey shows that the infection rate in Cyprus (32.3 %) is consistent with the recently reported prevalences in most areas in Europe, even though in some countries, such as Poland, France, Estonia and Albania, percentages of over 60.0% have been recorded [29,30,31,32,33]. During the last decade, the highest percentage of seropositive animals (84.7%) was recorded in the island of Majorca, in feral cats [34]. In areas of continental Spain, *T. gondii* seroprevalence was recorded as 36.9% in stray, 33.3% in farm and 25.5% in pet animals [35]. Interestingly, the lowest *T. gondii* seroprevalence in Europe was found in a breeding Persian cat colony, at 10.0% [36]. The latter indicates that cats living in a family household, without outdoor access, can still be infected, most probably by feeding of raw meat and less probably by ingesting infective oocysts that are mechanically transported in the house on the shoes or on arthropods [13]. Indeed, *T*. *gondii* is adapted to a tissue cyst-oral route of transmission in cats, as it has been shown that one tissue cyst is a competent infective dose, while 100 or more sporulated oocysts may be required to establish infection in a cat [37]. Accordingly, less than 50% of cats acquire a patent infection after ingesting oocysts, while almost all eliminate oocysts after tissue cyst consumption [22]. Similar to the lifestyle types evidenced as risk factors for *T. gondii* in previous surveys, the results of the present study show that cats with a likely lower veterinary care profile i.e., never vaccinated and without ectoparasite prevention, were significantly more likely to be seropositive. Furthermore, cats that had outdoor access, and cats that lived in a rural area, had a trend towards a statistically significant higher infection percentage than the cats with an indoor lifestyle or living in urban environment, respectively. This finding could be associated with the opportunity of a cat to prey on intermediate hosts (e.g., rodent or bird), which may occur more frequently for cats with outdoor access or living in rural areas. For these obvious reasons, outdoor access and living in peri-urban areas were also identified as risk factors for *T*. *gondii* infection in previous studies [13,35,38,39,40,41]. Accordingly, the association found in the present study between *Hepatozoon* spp. and *T*. *gondii* infection, using univariate analysis, is most likely related to the inadequate veterinary care and lack of ectoparasites prevention that usually occurs in cats with the above-mentioned lifestyle characteristics.

Age was not identified as a risk factor in the present study, despite there being a trend towards a significant association with seropositivity for cats more than one year of age compared to cats of one year of age or younger. Seroconversion in cats takes place 3–4 weeks post-infection and seropositivity lasts practically for all their lives, making age association an expected feature due to the accumulation of infection in older cats [31,42]. However, the fact that most cats are infected young, on their early hunting expeditions, abolish age-dependence after the first months of the animals’ life. Similarly, sex was also not associated with *T. gondii* seroprevalence in the present survey, as was also the case in several previous studies [42,43,44]. However, sex has been related with infection risk in some reports, with females found to be more prevalently infected than males [38], or with males being infected at a higher rate, but only when the studied cats had outdoor access [45]. Nevertheless, sex has an unclear relation to *T. gondii* infection prevalence.

In the present study, a significant association between FIV infection and seropositivity to *T*. *gondi* was found in the final multivariate logistic regression model. It has been suggested that immunosuppression associated with feline leukaemia virus (FeLV) and FIV infection may predispose cats to toxoplasmosis [46,47,48]. However, in some surveys, no association with these two retroviral infections has been documented [46,49]. The results of the present study are in accordance with most epizootiological surveys, which have evidenced that it is mainly FIV, rather that FeLV, that is correlated with *Toxoplasma* infection [41,42,50]. The immunodeficiency caused by FIV infection is most likely the underlying cause of cats’ enhanced susceptibility to *Toxoplasma* infection and for the antigenaemia and increased antibody synthesis resulted by the proliferation of cyst bradyzoites in already infected cats, facilitated by the impact of the viral infection [51,52].

CMhm infection showed a trend toward significance for *T*. *gondii* seropositivity but did not yield a significant association under multivariate logistic regression analysis. In accordance with the present results, in a survey in Albania, a higher percentage of cats were simultaneously infected with haemoplasmas and *T*. *gondii* than with haemoplasmas only, but the association was non-significant [31]. A possible association between common haemoplasmas infecting cats and susceptibility to *T*. *gondii* infection is worth further investigation in the future. 

*Toxoplasma gondii* represents an important agent in terms of disease and of zoonotic impact and economic implications in animal production. For these reasons, there is great merit in monitoring its epizootiological/epidemiological trends. In Cyprus, further studies in more animal species and an investigation of the factors involved in human infection are warranted to form a clearer image of infection prevalence and risk factors. Finally, raising public awareness of the ways to prevent *T*. *gondii* infection of animals and humans is pivotal in terms of the One Health concept.

## Figures and Tables

**Figure 1 pathogens-10-00882-f001:**
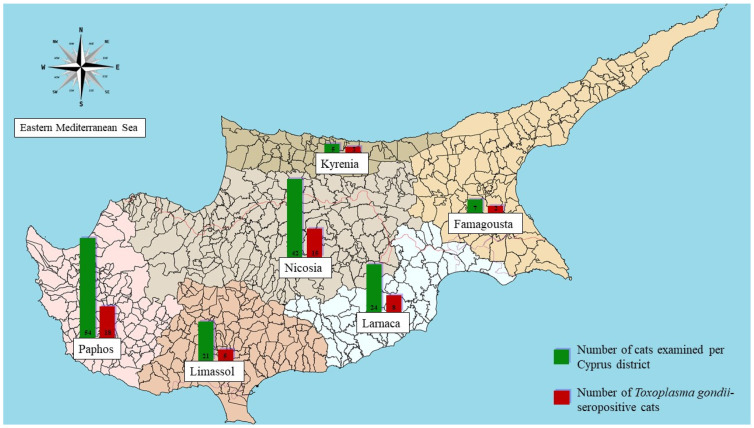
*Toxoplasma gondii* seropositivity in cats from the different districts of Cyprus.

**Table 1 pathogens-10-00882-t001:** Comparison of *Toxoplasma gondii* seropositivity and univariable analysis in cats from Cyprus per non-infectious categorical variable.

Variable (No. of Records)	No. of Cats (%)	*T*. *gondii* Seropositive Cats	
		No. (% ± 95% CI)	*p*-Values
**Age (155)**			
≤1 year	17 (11.0)	3 (17.6 ± 18.1)	*0.172*
>1 year (Ref.)	138 (89.0)	47 (34.1 ± 7.9)	
**Gender (155)**			
Male	81 (52.3)	25 (30.9 ± 10.0)	0.698
Female (Ref.)	74 (47.7)	26 (33.8 ± 10.7)	
**Breed (155)**			
Non-Pedigree	143 (92.3)	49 (34.3 ± 7.7)	*0.065*
Pedigree (Ref.)	12 (7.7)	1 (8.3 ± 15.6)	
**Housing (155)**			
Access to outdoors	119 (76.8)	43 (36.1 ± 12.9)	*0.061*
Indoors only (Ref.)	36 (23.2)	7 (19.4 ± 12.9)	
**Lifestyle (155)**			
Shelter-feral	32 (20.6)	13 (40.6 ± 17.0)	0.256
Owned (Ref.)	123 (79.4)	37 (30.1 ± 8.1)	
**District (155)**			
Paphos (Ref.)	54 (34.8)	18 (33.3 ± 12.6)	0.280
Nicosia	42 (27.1)	15 (35.7 ± 14.5)	
Larnaca	26 (16.8)	8 (30.8 ± 17.7)	
Limassol	21 (13.5)	5 (23.8 ± 18.2)	
Famagousta	7 (5.0)	2 (28.6 ± 33.5)	
Kyrenia	5 (3.2)	2 (40.0 ± 42.9)	
**Habitat (155)**			
Rural	59 (38.1)	23 (39.0 ± 12.4)	*0.160*
Urban (Ref.)	96 (61.9)	27 (28.1 ± 8.9)	
**Travel history (155)**			
Never travelled abroad	141 (91.0)	45 (31.9 ± 7.7)	0.772
Travelled abroad (Ref.)	14 (9.0)	5 (35.7 ± 25.0)	
**Health status (155)**			
Non-healthy	115 (74.2)	39 (33.9 ± 8.6)	0.455
Healthy (Ref.)	40 (25.8)	11 (27.5 ± 13.8)	
**Vaccination status (146)**			
Never vaccinated	41 (28.1)	19 (46.3 ± 15.2)	**0.016**
Vaccinated (Ref.)	105 (71.9)	27 (25.7 ± 8.3)	
**Ectoparasite prevention (146)**			
Never applied	56 (38.4)	24 (42.9 ± 12.9)	**0.020**
Applied (Ref.)	90 (61.6)	22 (22.4 ± 8.6)	
**Haematocrit (118)**			
Anaemic	28 (23.7)	11 (39.3 ± 18.0)	0.358
Non-anaemic	90 (76.3)	27 (30.0 ± 9.5)	

*p*-values < 0.2 but > 0.05 are shown in italics. Significant *p*-values ≤ 0.05 are shown in **bold**; CL: Confidence interval; Ref: Reference category.

**Table 2 pathogens-10-00882-t002:** Comparison of *Toxoplasma gondii* seropositivity and univariable analysis in cats from Cyprus with different co-infective agents.

Variable (No. of Records)	No. of Cats (%)	*T*. *gondii* Seropositive Cats	
		No. (% ± 95% CI)	*p*-Values
**FeLV antigen (152)**			
Positive	9 (5.9)	1 (11.1 ± 20.5)	*0.152*
Negative (Ref.)	143 (94.1)	49 (34.3 ± 7.8)	
**FIV antibody (152)**			
Positive	27 (17.8)	18 (66.7 ± 17.7)	**<0.0001**
Negative (Ref.)	125 (82.2)	32 (25.6 ± 7.6)	
**Any haemoplasma PCR (155)**			
Positive	39 (25.2)	16 (41.0 ± 15.4)	0.678
Negative (Ref.)	116 (73.8)	34 (29.3 ± 8.3)	
**Mhf PCR (155)**			
Positive	10 (6.5)	4 (40.0 ± 30.4)	0.588
Negative (Ref.)	145 (93.5)	46 (31.7 ± 7.6)	
**CMhm PCR (155)**			
Positive	31 (20.0)	14 (45.2 ± 17.5)	*0.086*
Negative (Ref.)	124 (80.0)	36 (29.0 ± 7.9)	
**CMt PCR (155)**			
Positive	9 (5.8)	3 (33.3 ± 30.8)	0.943
Negative (Ref.)	146 (94.2)	47 (32.2 ± 7.6)	
**Any FVBP PCR (155)**			
Positive	73 (47.1)	29 (39.7 ± 11.2)	*0.061*
Negative (Ref.)	82 (52.9)	21 (25.6 ± 9.4)	
***L*. *infantum* infection * (155)**			
Positive	12 (7.7)	3 (25.0 ± 24.5)	0.576
Negative (Ref.)	143 (92.3)	47 (32.9 ± 7.7)	
***B*. *henselae* PCR (155)**			
Positive	16 (10.3)	7 (43.8 ± 24.3)	0.576
Negative (Ref.)	139 (89.7)	43 (30.9 ± 7.7)	
***Hepatozoon* spp. PCR (155)**			
Positive	62 (40.0)	26 (41.9 ± 12.3)	**0.035**
Negative (Ref.)	93 (60.0)	24 (25.8 ± 8.9)	

*p*-values < 0.2 but > 0.05 are shown in italics. Significant *p*-values ≤ 0.05 are shown in **bold**; Any haemoplasma PCR: Positivity in at least one of the following haemoplasma PCRs: Mhf, CMhm and CMt; CI: Confidence interval; CMhm: “*Candidatus* Mycoplasma haemominutum”; CMt: “*Candidatus* Mycoplasma turicensis”; FeLV: feline leukaemia virus; FIV: feline immunodeficiency virus; FVBP: Feline Vector Borne Pathogen—positive for at least one of the PCRs for *B*. *henselae*, *Ehrlichia*/*Anaplasma* spp. and/or *Hepatozoon* spp., and/or *L*. *infantum* infection; Mhf: *Mycoplasma haemofelis*; Ref: Reference category; * *L*. *infantum* infection: confirmed by DNA sequencing following confirmatory qPCR and/or positive ELISA.

**Table 3 pathogens-10-00882-t003:** Multivariate logistic regression models regarding *Toxoplasma gondii* seropositivity in cats from Cyprus.

	Odds Ratio (95% CI)	*p*-Value
**FIV antibody**		
Positive	5.0 (2.0–13.0)	**0.001**
Negative	Ref.	
**Vaccination status**		
Never vaccinated	2.6 (1.2–5.7)	**0.019**
Vaccinated	Ref.	

CI: Confidence interval; Ref: Reference category; FIV: feline immunodeficiency virus; Significant *p*-values ≤ 0.05 are shown in **bold**.

## Data Availability

The Appendix A has all the datasets supporting the findings of this study.

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
