# Peer review of "Seroprevalence of Toxoplasma gondii in Cats from Cyprus"

_pathogens, 2021, doi:10.3390/pathogens10070882_

Round 1
Reviewer 1 Report
The article, " Seroprevalence of Toxoplasma gondii in cats from Cyprus" is a generally, well-structured and written paper that has summarized the results of the prevalence of the formerly mentioned disease with other variables, some which relate to lifestyle, age and sex, and others which relate to co-infection with other infectious diseases.
The authors have shown interesting results. However, I think that the article is better suited to a journal of regional importance like "Veterinary Parasitology: Regional studies and reports" as others have already explored much of this subject extensively.
Major comments:
Methodology:
More details should be mentioned as to how age was estimated in shelter cats. Also, since the paper discusses the T. gondii seroprevalence in cats, additional critical information of the tests is needed in this section as to how positivity was defined ie. antibody titer levels for positivity and specificity and sensitivity of tests according to the company.
The Discussion section needs to be better targeted concentrating more on the discussion of the results towards a take home message. Since the subject has been reported and discussed by others, it would be better to reduce the unnecessary redundancy here.
Minor comments:
Line 30: Arrange alphabetically
Line 48: Remove comma after "goat"
Line 54: Add comma after "time".
Lines 77-79: Omit the last sentence. This is rather a result and conclusion of the study and does not need to be included in the introduction.
Line 90: How was age estimated for shelter cats, especially the age estimate, ≤ 1 year and >1 year?
Lines 99: Please add the breakdown of antibody levels for positivity, negative and borderline definition (if relevant).
Line 105: What is CMhm? Not clear in the sentence?
Tables: I would advise to include Tables 3 and 4 in Tables 1 and 2, respectively. Please add the analyses and p - values in another column adjoining each variable. Also, it is recommended to remove the horizontal gridlines in the Tables. All abbreviations listed under each table need to be arranged alphabetically.
Table 2: Legend, change to "Table 2. Comparison of Toxoplasma gondii seropositivity in cats from Cyprus with different co-infective agents.
Line 200: Dogs, especially "puppies"…
Line 204: "causes of abortion, greatly affecting production"
Line 213: Adopt, "In Cyprus, T. gondii seroprevalence in humans has…."
Line 222: "main mode of human…"
Lines 225-229: In our present study, all districts of Cyprus displayed similar rates of feline infection, which is not surprising as there are no boundaries on the island that could isolate cat populations, and similarly the intermediate hosts playing a key role in feline infection (rodents and birds) are abundant in all districts, a…."
Line 232: "their population has reached 1.5 million"
Lines 277-278: "Nevertheless, sex has an unclear relation to T. gondii infection probability, as in other studies, female cats were more frequently infected than males [41]".
Author Response
The authors’ responses to the reviewers are shown italicised for clarity.
The authors would like to thank the reviewer for the positive comments and constructive feedback. All comments have been taken into consideration and the manuscript has been revised accordingly. Please see below the point-to-point replies.
Reviewer(s)' Comments to Author:
Reviewer: 1
The article, " Seroprevalence of Toxoplasma gondii in cats from Cyprus" is a generally, well-structured and written paper that has summarized the results of the prevalence of the formerly mentioned disease with other variables, some which relate to lifestyle, age and sex, and others which relate to co-infection with other infectious diseases.
The authors have shown interesting results. However, I think that the article is better suited to a journal of regional importance like "Veterinary Parasitology: Regional studies and reports" as others have already explored much of this subject extensively.
Major comments:
Methodology:
More details should be mentioned as to how age was estimated in shelter cats.
We thank the reviewer for these suggestions. The age was not estimated but recorded based on the medical records of each cat from the veterinary clinics/shelter (most animals arrived at the shelters as a few months old kittens) or on the basis of their teeth (deciduous or adult teeth, as well as the wear, tear and tartar buildup).. We have adreessed this by adding the following:
Line 90: (≤1 year or >1 year, on the basis of the records or teeth characteristics), sex…
Also, since the paper discusses the T. gondii seroprevalence in cats, additional critical information of the tests is needed in this section as to how positivity was defined ie. antibody titer levels for positivity and specificity and sensitivity of tests according to the company.
We thank the reviewer for highlighting this. We have expanded the material and methods section which now state the following:
Lines 101-107: The kit uses as substrate a purified peptide of the main P30 T. gondii protein and as conjugates a multi-species peroxidase (po), has shown a high level of concordance to other serological tests, and has been proved useful for toxoplasmosis serological screening in cats (Parigi et al 2012). The test was performed following the manufacturer’s instructions. Briefly, for each sample the sample to positive ration (S/P %) was calculated taking in account the optical101 density (OD) of the positive control (pc) and negative control (nc) as follows:
Samples with an S/P % ≤40% were considered negative, between 40% and 50% were considered doubtful and ≥50% were considered positive.
The Discussion section needs to be better targeted concentrating more on the discussion of the results towards a take home message. Since the subject has been reported and discussed by others, it would be better to reduce the unnecessary redundancy here.
We thank the reviewer for pointing this. We have removed a series of sentences in discussion. However, some general data, even though possibly known to the specialized readers, remain, to provide a more complete presentation of the topic.
Minor comments:
Line 30: Arrange alphabetically
Line 48: Remove comma after "goat"
Line 54: Add comma after "time".
We thank the reviewer for the above corrections which all have been made.
Lines 77-79: Omit the last sentence. This is rather a result and conclusion of the study and does not need to be included in the introduction.
The last years the vast majority of biomedical journals (e.g. Nature, Lancet) as well as Pathogens (“Finally, briefly mention the main aim of the work and highlight the principal conclusions”) do suggest including a compact sentence with the results of the paper in introduction. We also believe is useful for the reader.
Line 90: How was age estimated for shelter cats, especially the age estimate, ≤ 1 year and >1 year?
Please see response to your first point, line 90
Lines 99: Please add the breakdown of antibody levels for positivity, negative and borderline definition (if relevant).
Please see response to your second point, lines 101-107.
Line 105: What is CMhm? Not clear in the sentence?
We thank the reviewer for highlighting that this abbreviation was not explained. We have added the full name before the abbreviation:
Line 116: …for Bartonella henselae, “Candidatus Mycoplasma haemominutum” (CMhm),
Tables: I would advise to include Tables 3 and 4 in Tables 1 and 2, respectively. Please add the analyses and p - values in another column adjoining each variable. Also, it is recommended to remove the horizontal gridlines in the Tables. All abbreviations listed under each table need to be arranged alphabetically.
We thank the reviewer for this advice. We have now merged the tables following your suggestion, CI are included and we have listed the abbreviations alphabetically as recommended. See lines 146-171.
Table 2: Legend, change to "Table 2. Comparison of Toxoplasma gondii seropositivity in cats from Cyprus with different co-infective agents.
Line 200: Dogs, especially "puppies"…
Line 204: "causes of abortion, greatly affecting production"
Line 213: Adopt, "In Cyprus, T. gondii seroprevalence in humans has…."
Line 222: "main mode of human…"
Lines 225-229: In our present study, all districts of Cyprus displayed similar rates of feline infection, which is not surprising as there are no boundaries on the island that could isolate cat populations, and similarly the intermediate hosts playing a key role in feline infection (rodents and birds) are abundant in all districts, a…."
Line 232: "their population has reached 1.5 million"
Lines 277-278: "Nevertheless, sex has an unclear relation to T. gondii infection probability, as in other studies, female cats were more frequently infected than males [41]".
The authors would like to thank again the reviewer 1 for the detail correction of the manuscript. All the above changes have been made.
Reviewer 2 Report
The manuscript describes study on the seroprevalence of Toxoplasma gondii in cats from Cyprus.
The authors of the article provided valuable information from the field of parasitology which can be also interesting for epidemiologist.
However, I have some comments on the results part.
At first, the authors shows the seroprevalence of T. gondii among the multiple co-infecting pathogens in two tables (table 1 and table 2) with the same tittles. Please change the tittle of one of the table.
Secondly, the tables 3 and 4 contains the data which are very limited. Please consider including them for table 1 and 2.
Furthermore, it would be worth if the authors add 95% CI of the seroprevalence of T. gondii infection for each variable.
Author Response
The authors’ responses to the reviewers are shown italicised for clarity.
The authors would like to thank the reviewer for the positive comments and constructive feedback. All comments have been taken into consideration and the manuscript has been revised accordingly. Please see below the point-to-point replies.
Reviewer(s)' Comments to Author:
Reviewer: 2
The manuscript describes study on the seroprevalence of Toxoplasma gondii in cats from Cyprus.
The authors of the article provided valuable information from the field of parasitology which can be also interesting for epidemiologist.
However, I have some comments on the results part.
At first, the authors shows the seroprevalence of T. gondii among the multiple co-infecting pathogens in two tables (table 1 and table 2) with the same tittles. Please change the tittle of one of the table.
We thank the reviewer this comment. The title of Table 1 refers to the non-infectious categorical variable while this of Table 2 refers to the different co-infections. The title of Table 2 has been now slightly changed, according to the suggestion of Reviewer 1.
Secondly, the tables 3 and 4 contains the data which are very limited. Please consider including them for table 1 and 2. Furthermore, it would be worth if the authors add 95% CI of the seroprevalence of T. gondii infection for each variable.
We thank the reviewer for these comments. We have now merged the tables following this suggestion, CI are included and we have listed the abbreviations alphabetically as per Reviewer 1 recommendation. See lines 146-171.
Reviewer 3 Report
Dear authors, The manuscript was very well presented. I wonder if you, please answer some questions. 1. How the authors determined the sample size? 2. How the authors determined if the sample size is enough for a seroprevalence study? 3. Improve Figure 1 (change colors, use the same font, size, etc.). Please correct these small details. 4. Please bring us information about test sensibility and specificity and if the authors use any “gold standard” methodology to support data. 5. Please provide information about sensibility and specificity comparing data between all other techniques used to determine Toxoplasma seroprevalence Lanes: 156, 174, 175, 176, please use italics for scientific namesAuthor Response
The authors’ responses to the reviewers are shown italicised for clarity.
The authors would like to thank the reviewer for the positive comments and constructive feedback. All comments have been taken into consideration and the manuscript has been revised accordingly. Please see below the point-to-point replies.
Reviewer(s)' Comments to Author:
Reviewer: 3
Dear authors, The manuscript was very well presented. I wonder if you, please answer some questions.
- How the authors determined the sample size? 2. How the authors determined if the sample size is enough for a seroprevalence study?
As this is the first seroprevalence study for T. gondi in the Cypriot feline population there were data to be used for making a valid sample size selection and estimating actual number of cases need for a prevalence study. Furthermore, the samples derived from an already published study, thus act as samples of “convenience”, which is often the case in many veterinary publications. Hopefully this study will be a stepping stone for further toxoplasmosis studies in Cyprus.
- Improve Figure 1 (change colors, use the same font, size, etc.). Please correct these small details.
Following this advice, we changed Figure 1. Now all letters are in the same font (Times New Roman) and size 16. The original colours were retained as we believe they are quite clear, and all other combinations of colours tried gave a particularly “heavy” result.
- Please bring us information about test sensibility and specificity and if the authors use any “gold standard” methodology to support data. 5. Please provide information about sensibility and specificity comparing data between all other techniques used to determine Toxoplasma seroprevalence
We thank the reviewer this point. Following a similar comment from reviewer 1, we have expanded the M&M section with the appropriate information. Please see line 101-112
Lanes: 156, 174, 175, 176, please use italics for scientific names
Candidatus species have the particularity that the candidate scientific is not written in italics. Thus, the current format of writing the feline haemoplasmas is the appropriate one.
Round 2
Reviewer 1 Report
The authors have significantly improved on the article.
Minor comments
Please correct and remove the double fullstops in the first paragraph of the "Discussion".
Also, please attempt to organize the additional footnotes under each of the Tables better and place the Tables on one page instead of as is shown in the pdf format here.
Author Response
Reviewer(s)' Comments to Author:
Reviewer: 1
The authors have significantly improved on the article.
Minor comments
Please correct and remove the double fullstops in the first paragraph of the "Discussion". Also, please attempt to organize the additional footnotes under each of the Tables better and place the Tables on one page instead of as is shown in the pdf format here.
We thank the reviewer for these suggestions. All of them have been corrected in the revised manuscript.
More details should be mentioned as to how age was estimated in shelter cats.
We believe that the explanation provided after the first review process is as complete as it is part of standard veterinary practice to record the age of an animal, including the cats from the shelters, and being able to determine if a cat is more than a year old. Also we have re-checked our initial records from each case and all ages were recorded from the health records of each vet practise including the shelter cats.
Reviewer 3 Report
Dear Authors,
Thank you for taking the suggestions into account and adding the requested information. The manuscript has a significant improvement.
I have observed some minor typos, which are the following:
Lane 129: model. For both the univariate and multivariate model, a P-value ≤ 0.05
Lane 167: Univariate analysis (Tables 1 and 1)
Lane 185: Ref: rReference category
Lane 190: animals, both wild and domestic. . Clinical
Lane 281: ‘Candidatus Mycoplasma haemominutum’
This research work shows us the high seroprevalence of Toxoplasmosis, the possible side effects in infected people, and invites health authorities to develop health policies, establish better health strategies to reduce the prevalence of Toxoplasma gondii, and develop research to identify new anti-Toxoplasma compounds that lead to reduction or eradication of the parasite.
Author Response
The authors would like to thank the reviewer for the positive comments and additional constructive feedback. All comments have been taken into consideration and the manuscript has been revised accordingly. Please see below the point-to-point replies.
Reviewer(s)' Comments to Author:
Reviewer: 3
Dear Authors,
Thank you for taking the suggestions into account and adding the requested information. The manuscript has a significant improvement.
I have observed some minor typos, which are the following:
Lane 129: model. For both the univariate and multivariate model, a P-value ≤ 0.05
Lane 167: Univariate analysis (Tables 1 and 1)
Lane 185: Ref: rReference category
Lane 190: animals, both wild and domestic. . Clinical
Lane 281: ‘Candidatus Mycoplasma haemominutum’
We thank the reviewer for all of these suggestions and corrections. All of them have been corrected in the reviewed manuscript.